# Untargeted Metabolomics for the Diagnosis of Exocrine Pancreatic Insufficiency in Chronic Pancreatitis

**DOI:** 10.3390/medicina57090876

**Published:** 2021-08-26

**Authors:** Caridad Díaz, Cristina Jiménez-Luna, Carmelo Diéguez-Castillo, Ariadna Martín, José Prados, José Luis Martín-Ruíz, Olga Genilloud, Francisca Vicente, José Pérez del Palacio, Octavio Caba

**Affiliations:** 1Fundación MEDINA, Centro de Excelencia en Investigación de Medicamentos Innovadores, 18012 Granada, Spain; caridad.diaz@medinaandalucia.es (C.D.); ariadna.martin@medinaandalucia.es (A.M.); olga.genilloud@medinaandalucia.es (O.G.); francisca.vicente@medinaandalucia.es (F.V.); jose.perezdelpalacio@medinaandalucia.es (J.P.d.P.); 2Center of Biomedical Research (CIBM), Institute of Biopathology and Regenerative Medicine (IBIMER), University of Granada, 18012 Granada, Spain; crisjilu@ugr.es (C.J.-L.); ocaba@ugr.es (O.C.); 3Department of Gastroenterology, San Cecilio University Hospital, 18012 Granada, Spain; carmelo89dc@gmail.com (C.D.-C.); jlmartin@ugr.es (J.L.M.-R.)

**Keywords:** metabolomics, exocrine pancreatic insufficiency, chronic pancreatitis, biomarker, diagnosis, high-resolution mass spectrometry

## Abstract

*Background and Objectives*: The clinical manifestations and course of chronic pancreatitis (CP) are often nonspecific and variable, hampering diagnosis of the risk of exocrine pancreatic insufficiency (EPI). Development of new, reproducible, and non-invasive methods to diagnose EPI is therefore a major priority. The objective of this metabolomic study was to identify novel biomarkers associated with EPI. *Materials and Methods*: We analyzed 53 samples from patients with CP, 32 with and 21 without EPI, using an untargeted metabolomics workflow based on hydrophilic interaction chromatography coupled to high-resolution mass spectrometry. Principal component and partial least squares-discriminant analyses showed significant between-group differentiation, and univariate and multivariate analyses identified potential candidate metabolites that significantly differed between samples from CP patients with EPI and those without EPI. *Results*: Excellent results were obtained using a six-metabolic panel to diagnose the presence of EPI in CP patients (area under the ROC curve = 0.785). *Conclusions*: This study confirms the usefulness of metabolomics in this disease setting, allowing the identification of novel biomarkers to differentiate between the presence and absence of EPI in CP patients.

## 1. Introduction

Chronic pancreatitis (CP) is a progressive inflammatory disease characterized by irreversible morphological changes of the pancreatic gland, by fibrosis, and by impairment of exocrine and endocrine functions [1]. Worldwide, CP affects 0.4–5% of the population, reducing the life quality and expectancy of affected patients [2]. The diagnosis of CP is largely based on the presence of morphological or functional changes, but no standard diagnostic criteria or techniques have been established [3,4]. It is crucial to detect CP before its last stages, when there is a high risk of serious complications (e.g., diabetes mellitus, stenosis, pancreatic cancer, etc.) and the substantial restoration of pancreatic function becomes highly challenging [5].

Exocrine pancreatic insufficiency (EPI) consists of the destruction of the pancreatic parenchyma, with a loss of acinar cells and/or obstruction of pancreatic ducts, leading to an inadequate secretion of pancreatic enzymes and bicarbonate to maintain normal digestion [6]. CP is the most frequent cause of EPI in adults [7]. The diagnosis of EPI is currently based on maldigestion-related symptoms, nutritional markers, and noninvasive pancreatic function tests (e.g., fecal elastase-1 [FE-1]) [8]. Direct methods are more sensitive but are laborious, expensive, and invasive [9], whereas indirect methods are less costly, easier to perform, and non-invasive, but are only moderately sensitive for the early diagnosis of EPI [10,11].

The study of metabolism is an emerging and powerful tool for identifying novel biomarkers to achieve early diagnosis and/or evaluation of the course of different diseases [12]. Metabolites represent the end products of gene transcription and protein expression and therefore characterize the phenotype of an organism at a given time [13]. Hence, analysis of differences between normal and altered metabolic pathways may yield new biomarkers for diagnosis [14]. The aim of untargeted metabolomics is to analyze metabolites in an unbiased manner and has recently become widely applied to evaluate metabolic pathway changes associated with alterations [15]. High-resolution mass spectrometry (HRMS) is highly useful in untargeted metabolomics by combining the ability to determine masses within 3 ppm with the technology to investigate unknown compounds by collision-induced dissociation (or MS/MS), allowing exploration of relationships between parent ions and their fragments [16].

The objective of this metabolic study was to use hydrophilic interaction chromatography (HILIC) coupled to HRMS to identify potential biomarkers for the diagnosis of EPI in serum samples from CP individuals; a metabolite-based predictive model was developed to discriminate between patients with and without EPI.

## 2. Materials and Methods

### 2.1. Sample Collection

Our study included 53 samples from CP patients. Baseline characteristics are detailed in Table 1.

Alcoholic habits (defined by the intake of at least one standard drink unit of alcohol per day), sex, and cardiovascular events were evaluated using chi-squared distribution and found to be non-significant (*p* = 0.1478 for alcoholic habit, *p* = 0.659 for cardiovascular events, and *p* = 0.0102 for sex.

In order to achieve maximum reproducibility, the variation in parameters related to patients and sampling (fasting state, time of day of sampling, etc.) was minimized, using blood samples obtained from patients at San Cecilio University Hospital (Granada, Spain) between February and September 2017. The study was approved by the ethics committee of the hospital (approval code: 1269-M1-19), and all clinical investigations were conducted according to the principles of the Declaration of Helsinki. Written informed consent was obtained from all patients before their enrolment in the study.

Clotting of samples was activated by collecting blood from selected patients in BC vacutainer SSTII Advance tubes with silica (Becton Dickinson, Franklin Lakes, NJ, USA), centrifuging for 10 min at 2450 rpm, and aspirating the supernatant, which was stored at −80 °C.

### 2.2. Diagnosis of Exocrine Pancreatic Insufficiency

The exocrine pancreatic function of CP patients was assessed according to FE-1 levels measured with the BioServ Diagnostics Pancreatic Elastase ELISA kit, following the manufacturer’s instructions (BioServ Analytics and Medical Devices Ltd., Rostock, Germany) and considering FE-1 ≥ 200 μg/g as normal pancreatic function, FE-1 = 100–200 μg/g as mild-moderate EPI, and FE-1 < 100 μg/g as severe EPI.

### 2.3. Metabolite Extraction

AcN (1:8 sample/AcN) was added to serum samples and shaken for 2 min to remove proteins. After centrifugation at 15,200 rpm for 10 min at 4 °C, supernatants were collected and transferred to vials and then evaporated using the GeneVac HT-8 evaporator (Savant, Holbrook, NY, USA). AcN/water (50:50) with 0.1% formic acid was used to reconstitute obtained dry residues, shaking for 1 min. All samples were kept at 4 °C throughout analytical procedures.

### 2.4. HILIC-HRMS Analysis

Chromatographic separation was performed as follows: 0.00–0.10 min 99% eluent B, 0.10–7.00 min 30% eluent B, 7.00–7.10 min 99% eluent B, and 7.10–10.00 min 99% eluent B, using a Waters Xbridge BEH amide column (2.1 × 150 mm, 2.5 µm). Column temperature was maintained at 45 ºC. The elution flow rate was 400 µL/min. Agilent Series 1290 was used as LC instrument.

Mass detection was carried out using an SCIEX Triple TOF 5600 quadrupole-time-of-flight mass spectrometer in ESI (-) mode (SCIEX, Concord, ON, Canada) for HILIC analysis. Fragmentation and mass spectra were obtained by operating the TripleTOF 5600 using a TOF method and an information-dependent acquisition (IDA) technique to simultaneously collect full-scan HRMS and MS/MS information.

The IDA technique was used to fragment the eight most intense ions. Exact mass calibration was automatically performed every eight injections. An MP sample was run every 50 samples to identify impurities from the solvents or extraction procedure and to test carryover contamination from intense peaks. A QC sample was injected every 10 chromatographic runs to check variability in the analysis.

### 2.5. Data Set Creation

The retention time (RT) and mass/charge (*m*/*z*) were evaluated to study the analytical reproducibility using PeakView software (version 1.0 with Formula Finder plug-in version 1.0, AB SCIEX, Concord, ON, Canada). Raw data were processed with MarkerView software (version 1.2.1, AB SCIEX, Concord, ON, Canada), which performs peak detection, alignment, and data filtering.

Peak detection was carried out using an algorithm in the RT range of 1–8 min and removing background noise (50 cps). Peak alignment was achieved using RT and *m*/*z* tolerances of 0.15 min and 15 ppm, respectively. Next, a filter was applied by “presence” to retain masses appearing in at least 5 samples in the study case groups, only considering monoisotopic peaks. Finally, a filtering procedure with *t*-test (*p* < 0.05) and fold-change (>2) was employed to identify differentially expressed mass signals between MP samples and case study samples.

### 2.6. Analytical Validation

Metaboanalyst 3.0 Web Server was used to transform the data matrix into a Gaussian-type distribution by total area sums, auto scaling, and cube transformation. The study was evaluated by the non-supervised PCA method. Variables with unacceptable reproducibility (RSD > 30%) were excluded. In the statistical validation, *R*^2^ and *Q*^2^ explained the goodness of fit and prediction, respectively. PLS-DA allowed the identification of outliers.

### 2.7. Statistical Analysis

Univariate analysis was performed with the Wilcoxon-test (*p* < 0.05), determining the statistical significance of between-group differences. Multivariate analysis was also conducted, carrying out PCA and a PLS-DA to identify *m*/*z* values responsible for the separation between groups. Finally, the fold change (>1.2) between groups was considered as election criterion for variability in the amounts of metabolites.

### 2.8. Biomarker Identification

Selected molecular components were identified by searching in Metlin, Human Metabolome Database, KEGG, and Lipid Maps databases, using a mass tolerance of <5 ppm. Next, PeakView software (version 1.0 with Formula Finder plug-in version 1.0, AB SCIEX, Concord, ON, Canada) was used to determine molecular formulas. *m*/*z* with the same molecular formula in experimental and database estimations were further analyzed by studying their experimental MS/MS spectra with information provided by the MassBank, NIST 2014 MS/MS, Human Metabolome Database, and Metlin databases.

### 2.9. Biomarker Evaluation

The sensitivity and specificity of the proposed biomarkers were tested using the receiver operator characteristic (ROC) curve, a non-parametric measure of biomarker utility. Univariate and multivariate ROC analyses were performed to evaluate the clinical value of the candidates as biomarkers individually and in combination with others.

## 3. Results

EPI was diagnosed or ruled out in CP patients using the FE-1 test: 60.4% of patients with CP patients were diagnosed with EPI (FE-1 < 200 µg/g) and the remaining 39.6% were not (NO-EPI). Among those diagnosed with EPI, 87.5% had severe EPI (FE-1 < 100 µg/g).

After alignment and filtering processes, a data matrix of 1262 metabolite features was obtained, 302 of which were monoisotopic. Among these, 254 features were differentially expressed in case samples (EPI and NO-EPI) and MP samples. After normalization, six of these variables were discarded due to unacceptable variability (RSD > 30%). The PCA score plot for the remaining 248 variables (Figure 1A) revealed a close clustering of the QC samples, indicating that the separation observed between EPI and NO-EPI was mainly due to biological factors, and the PLS-DA score plot (Figure 1B) suggested that it was feasible to discriminate between these patients. Three outliers were detected and excluded from the analysis. The *Q*^2^ value for the predictive ability of the PLS-DA model to discriminate between EPI and NO-EPI was 0.15 and the *R*^2^ value was 0.30, with a good separation between the groups in the PLS-DA model. Out of the total variance, 63.7% was explained by two components alone (Figure 1B).

Out of these 248 variables, 12 had a *p*-value < 0.05. Fold-change calculations (higher than 1.2) showed that 11 metabolites were regulated in opposite directions in patients with versus without EPI. These metabolites were tentatively identified, but only 7 of them could be matched in the database used (Table 2). The remaining significant metabolites could not be assigned an identity (*m*/*z* 303.9204 at 3.9 min, 841.9492 at 3.0 min, 977.9249 at 3.1 min and 1045.915 at 3.2 min). In all cases, these molecules were increased in EPI patients. Four phospholipids classified as phosphatidylserines (PS) and one phosphatidylcholine (PC) were increased in the EPI group, together with a peptide formed with arginine, threonine, and proline, although their position is unknown, and pentasine.

The potential of the candidate biomarkers was further evaluated by constructing univariate ROC curves to obtain the area under the ROC curve (AUC). Two metabolites showed an AUC > 0.75 (Figure 2). Two phosphatidylserines were the best individual biomarkers: PS (R1COOH + R2COOH = 41:4), with an AUC of 0.78 (95% CI 0.63–0.91); and PS (R1COOH + R2COOH = 39:2), with an AUC of 0.75 (95% CI 0.63–0.91).

We applied a random forest model to combine arginine–proline–threonine tripeptide, PC (16:0/5:0(OH)), pentasine, PS (R1COOH + R2COOH = 39:2), PS (R1COOH + R2COOH = 41:4), and PS (R1COOH + R2COOH = 43:6, obtaining an AUC of 0.79 (95% CI 0.62–0.92) for the combined set (Figure 3A), higher than the AUC for each independent biomarker. The confusion matrix correctly classified 22 patients with pancreatic insufficiency and misclassified 5, and it correctly classified 15 patients without pancreatic insufficient and misclassified 8 (Figure 3B).

## 4. Discussion

The early diagnosis of EPI in patients with CP is vital due to the increased risk of malnutrition and cardiovascular events [17,18]. Indirect methods are widely used but have a low sensitivity for early stages of EPI [8]. In the present study, a six-metabolite panel proved useful to discriminate between the presence and absence of EPI in patients with CP. One of the first metabolic studies for this purpose used high-pressure liquid-chromatography (LC) to assess the diagnostic capability of the bentiromide test to determine the concentration of p-aminobenzoic acid and its metabolites, and it detected a lower concentration of amines in patients with EPI than in controls. They concluded that this noninvasive method of evaluating pancreatic function could play an important role in the diagnosis of EPI [19].

Among the six metabolites identified as EPI biomarkers in the present study, three of them were phosphatidylserines (PS). These molecules are essential components of eukaryotic cells, present on the inner leaflet of their membranes and with an important role in apoptosis and blood clotting [20]. Interestingly, their levels on the surface of pancreatic cancer cells appear to be higher than on the surface of other types of cancer cell [21], and this abundance has led to proposals for the targeting of PS in combination with gemcitabine in the treatment of pancreatic cancer [22]. In the present study, these molecules were found at higher concentrations in EPI patients, in agreement with previous findings that the PS receptor on pancreatic cells is markedly upregulated in CP, alongside observations implicating pro-apoptotic genes, which are known to play a major role in the destruction underlying EPI [23].

In addition, one phosphatidylcholine (PC) was found at higher concentrations in sera from the EPI group versus NO-EPI group. PC is the major phospholipid component of cell membranes, more commonly found in the outer leaflet, and contributes to proliferative growth and programmed cell death [24]. Zeman et al. analyzed fatty acid (FA) profiles in plasma phosphatidylcholines from 96 CP patients and 108 controls; they found decreased fat intake and intrinsic changes in FA metabolism in the CP group due to the alteration of delta desaturase activities, resulting in higher monounsaturated FA and lower polyunsaturated FA contents [25].

The highest fold change between studied groups was observed in a peptide formed by arginine, threonine, and proline. Although their position in the peptide is unknown, a search was conducted of a peptide BLAST with all possible tripeptides, finding that arginine–proline–threonine and proline–arginine–threonine are constituent tripeptides of different types of integrin. These proteins have been associated with extracellular matrix destruction processes in inflammatory processes such as pancreatitis [26]. Moreover, it has been demonstrated that the intraperitoneal administration of arginine can induce selective and dose-dependent experimental pancreatitis in rats [27]. The action mechanism of arginine-induced pancreatitis was reported to be mediated by oxidative stress, inflammation, and apoptosis, among other processes [28]. High concentrations of proline have also been related to pancreatic alterations. In 2016, in vitro studies demonstrated that high doses of this cyclic amino acid cause toxicity and beta-cell dysfunction mediated by a decrease in insulin transcription and impairment of mitochondrial oxidative phosphorylation [29].

Finally, our panel also includes pentasine, a carboxylic acid with five carboxyl groups that belongs to the class of organic compounds known as pentacarboxylic acids and derivatives. Several studies have pointed to pentasine as an elastin cross-linking amino acid, which may be related to reduced elastin secretion [30]. Degradation of extracellular matrix proteins such as elastin is considered a prerequisite for the progression of various diseases, including tumor metastases [31]. In addition, extracellular matrices containing elastin-like polypeptide were found to improve islet transplantation outcomes in diabetic mice [32].

The proposed metabolites were identified by using the reported databases, and confirmation of their identity against biological standards would have strengthened the study. A further potential limitation is the lesser representation of females in the study population, attributable to the lower prevalence of CP among females and in part explained by their less frequent exposure to etiological factors [33]. In-depth studies are currently under way to test the capacity of this six-metabolite panel to discriminate between CP patients with and without EPI in wider samples containing higher percentages of patients with early stage EPI.

## 5. Conclusions

Blood samples from patients with chronic pancreatitis studied with hydrophilic interaction chromatography coupled to mass spectrometry yielded a panel of six metabolites that showed differential expression according to the presence or absence of exocrine pancreatic insufficiency. These results confirm the potential of metabolic studies to improve understanding of the pathogenesis of exocrine pancreatic insufficiency and to identify new diagnostic markers.

## 6. Patents

The patent PCT/EP2020/067255 results from the work reported in this manuscript.

## Figures and Tables

**Figure 1 medicina-57-00876-f001:**
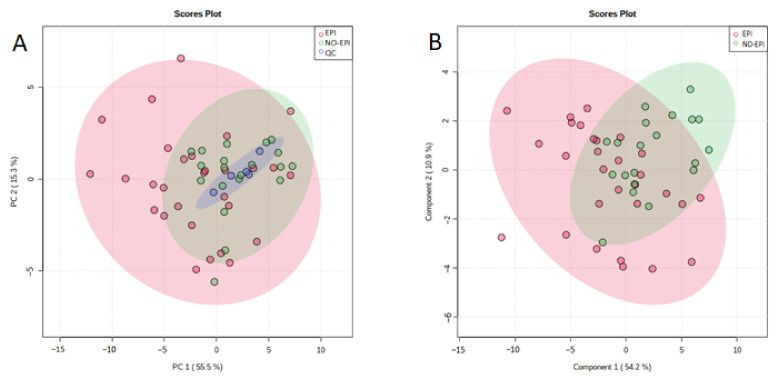
(**A**) Principal component analysis score plot; (**B**) partial least squares-discriminant analysis score plot. Pink dots correspond to EPI patients, green dots to NO-EPI patients, and blue dots to quality control samples.

**Figure 2 medicina-57-00876-f002:**
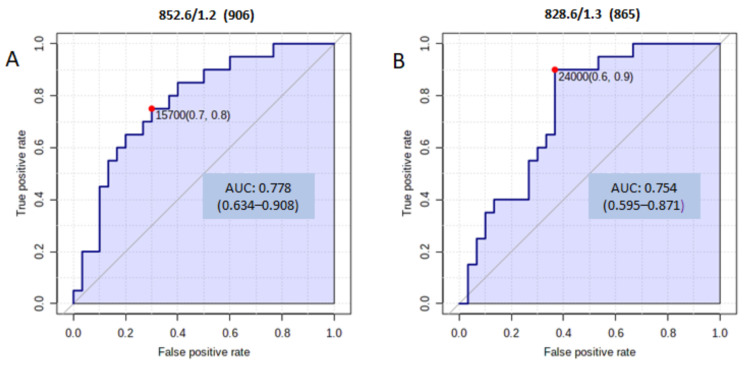
(**A**) AUC for PS (R1COOH + R2COOH = 41:4); (**B**) AUC for PS (R1COOH + R2COOH = 39:2).

**Figure 3 medicina-57-00876-f003:**
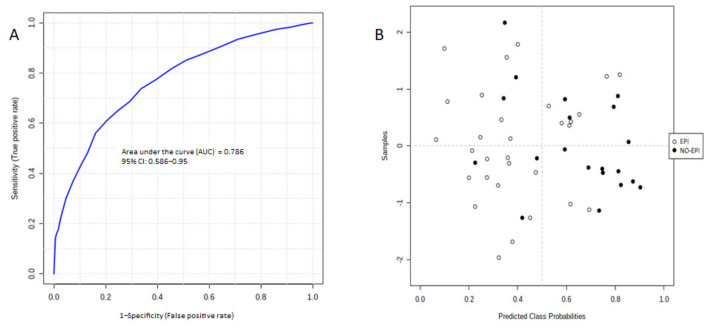
(**A**) ROC curves for the proposed six-metabolite panel; (**B**) classification using predicted group probabilities.

**Table 1 medicina-57-00876-t001:** Baseline characteristics of patients with chronic pancreatitis.

Characteristic	CP Patients (%)	EPI (%)	NO-EPI (%)
Number	53	32 (60.4)	21 (39.6)
Age (years)	55.7	56.7	54.3
* Sex			
Male	44 (83)	30 (93.7)	14 (66.7)
Female	9 (17)	2 (6.3)	7 (33.3)
Diagnostic test			
Endoscopic ultrasonography	38 (71.7)	21 (65.6)	17 (81)
Computed Tomography	10 (18.9)	6 (18.7)	4 (19)
Abdominal ultrasound	3 (5.6)	3 (9.4)	0 (0)
Anatomopathological study	2 (3.8)	2 (6.3)	0 (0)
* Alcoholic habit			
Yes	34 (64.2)	23 (71.9)	11 (52.4)
No	19 (35.8)	9 (28.1)	10 (47.6)
Smoking habit			
Yes	34 (64.2)	27 (84.4)	7 (33.3)
No	19 (35.8)	5 (15.6)	14 (66.7)
Body Mass Index			
Overweight/Obesity (>25)	24 (45.3)	10 (31.3)	14 (66.7)
Normal weight (18 > BMI < 25)	28 (52.8)	21 (65.6)	7 (33.3)
Underweight (<18)	1 (1.9)	1 (3.1)	0 (0)
* Cardiovascular events			
Yes	4 (7.6)	2 (6.3)	2 (9.5)
No	49 (92.4)	30 (93.7)	19 (90.5)
Abdominal pain			
Yes	24 (45.3)	16 (50)	8 (38.1)
No	29 (54.7)	16 (50)	13 (61.9)
Diarrhea			
Yes	12 (22.6)	10 (31.3)	2 (9.5)
No	41 (77.4)	22 (68.7)	19 (90.5)
* Diabetes			
Yes	26 (49.1)	20 (62.5)	6 (28.6)
No	27 (50.9)	12 (37.5)	15 (71.4)
Complications ^1^			
Yes	31 (58.5)	19 (59.4)	12 (57.1)
No	22 (41.5)	13 (40.6)	9 (42.9)
Hospital readmissions ^2^			
Yes	21 (39.6)	13 (40.6)	8 (38.1)
No	32 (60.4)	19 (59.4)	13 (61.9)
Treatment			
Endoscopic	8 (15.1)	6 (18.7)	2 (9.5)
Surgical	13 (24.5)	10 (31.3)	3 (14.3)
Analgesic	30 (56.6)	21 (65.6)	9 (42.9)
Substitutive enzyme	29 (54.7)	22 (68.7)	7 (33.3)
Oral antidiabetics	26 (49.1)	20 (62.5)	6 (28.6)
Statins/Fibrates	16 (30.2)	11 (34.4)	5 (23.8)

^1^ Compressive (pseudocyst or abscess), stenotic (biliary o duodenal), or vascular (pseudoaneurysm or splenoportal thrombosis); ^2^ Hospital admissions in a follow-up period of 12 months. * Chi square distribution *p* value > 0.01.

**Table 2 medicina-57-00876-t002:** Detailed information of the potential biomarkers of exocrine pancreatic insufficiency.

*m/z*	RT (min)	Tentative Identification	Δppm	Adduct	Fold Change
417.2103	1	Arginine-threonine-proline	0	+F.A.−H	1.74
610.3721	1.3	PC (16:0/5:0(OH))	1	−H	1.43
634.3334	2.1	Pentasine	1	−H−H_2_O	1.29
828.5726	1.3	PS (R1COOH + R2COOH = 39:2)	4	−H	1.21
852.5767	1.2	PS (R1COOH + R2COOH = 41:4)	1	H	1.24
876.5739	1.2	PS (R1COOH + R2COOH = 43:6)	2	H	1.39

Biomarkers were selected according to *t*-test (*p* < 0.05) and fold change (higher than 1.2) results. RT: Retention time. Fold change expressed as the ratio of the two averages (EPI/NO-EPI).

## Data Availability

The data of the study are present at the Fundacion MEDINA. The data will be available upon reasonable request.

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
