# Peer review of "Untargeted Metabolomics for the Diagnosis of Exocrine Pancreatic Insufficiency in Chronic Pancreatitis"

_medicina, 2021, doi:10.3390/medicina57090876_

Round 1

Reviewer 1 Report

Chronic pancreatitis and resulting exocrine pancreatic insufficency present an underrecognized diagnostic and treatment challenge. The authors present a novel way of identifying potential biomarkers using spectrometry and chromatography and provide potentially exciting results on a good sample population. There are however limitations to using these methods with which readers might not be familiar with and the discussion would benefit from highlighting this, as only strengths of the study are mentioned. In summary:

  • study limitations due to methodology and low representation of female population should be reported in the discussion
  • smoking status and presence of diabetes or insulin use in patients is not mentioned
  • Table 1 could be more readable with the inclusion of precentages (%) next to (n) of patients

Author Response

Comments and Suggestions for Authors:

Chronic pancreatitis and resulting exocrine pancreatic insufficency present an underrecognized diagnostic and treatment challenge. The authors present a novel way of identifying potential biomarkers using spectrometry and chromatography and provide potentially exciting results on a good sample population. There are however limitations to using these methods with which readers might not be familiar with and the discussion would benefit from highlighting this, as only strengths of the study are mentioned. In summary:

Point 1: Study limitations due to methodology and low representation of female population should be reported in the discussion.

Response 1: We have now included the following new paragraph on these limitations of our study:

“The proposed metabolites were identified by using the reported databases, and confirmation of their identity against biological standards would have strengthened the study. A further potential limitation is the lesser representation of females in the study population, attributable to the lower prevalence of CP among females and in part explained by their less frequent exposure to etiological factors [34].” (Page 8; lines 15-19).

Point 2: Smoking status and presence of diabetes or insulin use in patients is not mentioned.

Response 2: These data have now been included in Table 1.

Point 3: Table 1 could be more readable with the inclusion of percentages (%) next to (n) of patients.

Response 3: This has been done.

We are grateful for the comments and suggestions of the reviewer.

Reviewer 2 Report

The authors report the results of an untargeted metabolomics screen for novel biomarkers to separate an intention to diagnose population consisting of chronic pancreatitis (CP) patients with and without exocrine pancreatic insufficiency (EPI). The study consisted of 53 CP patients with 32 having EPI and 21 not. Serum samples were collected under similar conditions across the population and analyzed by LC-MS with information-dependent MS/MS to create fragments to further aid in the identification of molecular features.

After applying a reasonable set of filters to the dataset, the authors were able to identify 12 molecular features out of 248 that displayed statistically significant differences between the EPI and no EPI groups. Of those 12, 7 were able to be identified at a Tier 2 (PMID 24039616) level by matching to a external reference databases from which they were able to build a diagnostic panel with reasonable performance for a 6 metabolite Random Forest model with an AUC of 0.79.

This is an interesting discovery study with promising initial results. The authors are particularly interested in being able to correctly identify early-stage EPI patients where but there were only 4 early stage EPI subject out of the 32 and the assessment of EPI was performed by one of the indirect methods (FE-1) they describe as, "only moderately sensitive for the early detection of EPI." These aspects of the study should be discussed as limitations and thoughts on follow-up studies such as trying to use the 6 metabolite panel (either with or without FE-1) in comparison with a gold standard method like secretin enhanced MRCP.

Comments:

  1. Please briefly discuss the limitations of the study and how it might be possible to assess the performance of the 6 metabolite panel against a population containing a greater percentage of early-stage EPI patients.
  2. (Minor) Please relabel the Figure 1 legends to EPI and No EPI rather than IPE and No IPE to be consistent with the text.
  3. The narrative about the effects of individual amino acids (Arg, Pro, Thr) is not compelling as the real discussion should be centered around the 6 possible tripeptides (RPT, RTP, PRT, PTR, TRP, TPR) by searching a peptide BLAST (e.g., http://pepbank.mgh.harvard.edu/search/blast) for potential hits connected to the pancreas.
  4. Pentasine is an intriguing finding. Three hits for pentasine in PubMed (PMID: 2952445, 2490266, and 10992159) point to it being an elastin cross-linking amino acid. This seems like it could be a real finding related to reduced elastin secretion and should be briefly discussed.
  5. Please state which 6 metabolites were used for the Random Forest model.

Author Response

Comments and Suggestions for Authors:

The authors report the results of an untargeted metabolomics screen for novel biomarkers to separate an intention to diagnose population consisting of chronic pancreatitis (CP) patients with and without exocrine pancreatic insufficiency (EPI). The study consisted of 53 CP patients with 32 having EPI and 21 not. Serum samples were collected under similar conditions across the population and analyzed by LC-MS with information-dependent MS/MS to create fragments to further aid in the identification of molecular features.

After applying a reasonable set of filters to the dataset, the authors were able to identify 12 molecular features out of 248 that displayed statistically significant differences between the EPI and no EPI groups. Of those 12, 7 were able to be identified at a Tier 2 (PMID 24039616) level by matching to a external reference databases from which they were able to build a diagnostic panel with reasonable performance for a 6 metabolite Random Forest model with an AUC of 0.79.

This is an interesting discovery study with promising initial results. The authors are particularly interested in being able to correctly identify early-stage EPI patients where but there were only 4 early stage EPI subject out of the 32 and the assessment of EPI was performed by one of the indirect methods (FE-1) they describe as, "only moderately sensitive for the early detection of EPI." These aspects of the study should be discussed as limitations and thoughts on follow-up studies such as trying to use the 6 metabolite panel (either with or without FE-1) in comparison with a gold standard method like secretin enhanced MRCP. Comments:

Point 1: Please briefly discuss the limitations of the study and how it might be possible to assess the performance of the 6 metabolite panel against a population containing a greater percentage of early-stage EPI patients.

Response 1: As requested, we have added the following paragraph:

“In-depth studies are currently under way to test the capacity of this six-metabolite panel to discriminate between CP patients with and without EPI in wider samples containing higher percentages of patients with early-stage EPI.” (Page 8; lines 19-22).

Point 2: Please relabel the Figure 1 legends to EPI and No EPI rather than IPE and No IPE to be consistent with the text.

Response 2: Figure 1 legends have been relabeled accordingly.

Point 3: The narrative about the effects of individual amino acids (Arg, Pro, Thr) is not compelling as the real discussion should be centered around the 6 possible tripeptides (RPT, RTP, PRT, PTR, TRP, TPR) by searching a peptide BLAST (e.g., http://pepbank.mgh.harvard.edu/search/blast) for potential hits connected to the pancreas.

Response 3: We carried out this search, as now reported in the revised Discussion (Page 7; lines 34-38):

Although their position in the peptide is unknown, a search was conducted of a peptide BLAST with all possible tripeptides, finding that arginine-proline-threonine and proline-arginine-threonine are constituent tripeptides of different types of integrin. These proteins have been associated with extracellular matrix destruction processes in inflammatory processes such as pancreatitis [27].

Point 4: Pentasine is an intriguing finding. Three hits for pentasine in PubMed (PMID: 2952445, 2490266, and 10992159) point to it being an elastin cross-linking amino acid. This seems like it could be a real finding related to reduced elastin secretion and should be briefly discussed.

Response 4: We are grateful for this comment. We now discuss this finding in the following addition to the Discussion:

“Several studies have pointed to pentasine as an elastin cross-linking amino acid, which may be related to reduced elastin secretion [31]. Degradation of extracellular matrix proteins such as elastin is considered a prerequisite for the progression of various diseases, including tumor metastases [32]. In addition, extracellular matrices containing elastin-like polypeptide were found to improve islet transplantation outcomes in diabetic mice [33].” (Page 8; lines 9-14).

Point 5: Please state which 6 metabolites were used for the Random Forest model.

Response 5: As now reported in the Results section (Page 6; lines 24-26), the six proposed metabolites used for the Random Forest model were: arginine-proline-threonine tripeptide, PC(16:0/5:0(OH)), pentasine, PS (R1COOH+R2COOH=39:2), PS (R1COOH+R2COOH=41:4), and PS (R1COOH+R2COOH=43:6).

We are grateful for the comments and suggestions of the reviewer.